# Enrichment Methods for Murine Liver Non-Parenchymal Cells Differentially Affect Their Immunophenotype and Responsiveness towards Stimulation

**DOI:** 10.3390/ijms23126543

**Published:** 2022-06-11

**Authors:** Carolina Medina-Montano, Maximiliano Luis Cacicedo, Malin Svensson, Maria Jose Limeres, Yanira Zeyn, Jean Emiro Chaves-Giraldo, Nadine Röhrig, Stephan Grabbe, Stephan Gehring, Matthias Bros

**Affiliations:** 1Department of Dermatology, University Medical Center of the Johannes Gutenberg University Mainz, Langenbeckstraße 1, 55131 Mainz, Germany; gmedinam@uni-mainz.de (C.M.-M.); yanira.zeyn@uni-mainz.de (Y.Z.); jchavesg@students.uni-mainz.de (J.E.C.-G.); n.roehrig@uni-mainz.de (N.R.); stephan.grabbe@unimedizin-mainz.de (S.G.); 2Children’s Hospital, University Medical Center of the Johannes Gutenberg University Mainz, Langenbeckstraße 1, 55131 Mainz, Germany; mcaciced@uni-mainz.de (M.L.C.); malin.svensson@uni-mainz.de (M.S.); mj.limeres@uni-mainz.de (M.J.L.); stephan.gehring@uni-mainz.de (S.G.)

**Keywords:** liver perfusion, liver dissociation, liver sinusoidal endothelial cells, Kupffer cells, dendritic cells, adjuvant, toll-like receptor ligand, liposome

## Abstract

Hepatocytes comprise the majority of the liver and largely exert metabolic functions, whereas non-parenchymal cells (NPCs)—comprising Kupffer cells, dendritic cells and liver sinusoidal endothelial cells—control the immunological state within this organ. Here, we compared the suitability of two isolation methods for murine liver NPCs. Liver perfusion (LP) with collagenase/DNase I applied via the portal vein leads to efficient liver digestion, whereas the modified liver dissociation (LD) method combines mechanical dissociation of the retrieved organ with enzymatic degradation of the extracellular matrix. In cases of both LP and LD, NPCs were enriched by subsequent gradient density centrifugation. Our results indicate that LP and LD are largely comparable with regards to the yield, purity, and composition of liver NPCs. However, LD-enriched liver NPCs displayed a higher degree of activation after overnight cultivation, and accordingly were less responsive towards stimulation with toll-like receptor ligands that are frequently used as adjuvants, e.g., in nano-vaccines. We conclude that LP is more suitable for obtaining liver NPCs for subsequent in vitro studies, whereas LD as the less laborious method, is more convenient for parallel isolation of larger numbers of samples for ex vivo analysis.

## 1. Introduction

The liver is a central metabolic organ, but also plays an important immunological role, serving to maintain tolerance under homeostatic conditions [1,2,3]. The liver is comprised of hepatocytes, which make up approximately two thirds of its total cell population and exert largely metabolic functions, comprising the synthesis of numerous types of plasma proteins—for example, albumin [4], fatty acids [5], carbohydrates [6], and bile acid metabolism [7]—but are also pivotal for detoxification [8]. The heterogeneous group of non-parenchymal cells (NPCs) plays an important role in immune regulation [9]. Liver NPCs comprise liver sinusoidal endothelial cells (LSECs, approximately 50% of NPCs), Kupffer cells (KCs, 20%) that largely constitute the resident macrophage population, and dendritic cells (DCs, 25%). The minor fraction of liver NPCs is mainly composed of innate immune cells comprising natural killer (NK) cells and NKT cells [10], biliary epithelial cells [7], and hepatic stellate cells (HSC) [11]. The latter are located in the perisinusoidal region between the sinoids and hepatocytes, and under steady state conditions remain in a quiescent state, but play an important role in the induction of liver diseases [12]. HSC are activated in response to liver damage and in this state constitute the main producers of extracellular matrix factors—thereby promoting liver fibrosis [13]. Furthermore, chronic inflammation of a fibrotic liver enhances the likeliness of hepatocellular carcinoma (HCC) development [14]. HCC in turn reprograms HSC to differentiate towards cancer-associated fibroblasts, which besides other cell types such as tumor-associated macrophages, shape the immune-evasive tumor microenvironment [15]. With regard to the major liver NPC fractions, KCs were recognized early on as promoting liver diseases [16] such as nonalcoholic fatty liver disease [17] and alcoholic hepatitis [18]. In general, KCs have been attributed protolerogenic functions [19,20]. Similar to KCs, liver DCs have been reported to exert protolerogenic functions under steady state conditions [21], but have also been reported to display a stimulated, proimmunogenic state at later stages of NAFLD and to contribute to liver fibrosis [22].

LSECs form the lining of hepatic sinusoids and are characterized by numerous pore-like structures, termed fenestrae, which allow the rapid transport of, e.g., lipoproteins from hepatocytes into the sinusoidal blood stream [23]. Furthermore, LSECs release vasodilatory mediators upon shear stress, and impaired generation of these factors has been implicated in non-alcoholic fatty liver disease [24]. Moreover, LSECs express numerous types of endocytic scavenger and C-type lectin receptors [25] as well as the Fcγ receptor IIb (CD32) [26], which enable the uptake of different types of pathogens [27,28] and immune complexes [26] and promote immune homeostasis by inducing regulatory T cells [29].

Altogether, liver NPCs on the one hand have the common property of maintaining antigen-specific T cell tolerance under homeostatic conditions [30,31], but on the other hand, may contribute to NAFLD and other liver diseases in cases of dysfunction [24,32,33]. Liver NPCs have also come into the focus of nanodrug-associated research as a large fraction of systemically applied nano-formulations accumulates in the liver [34,35,36]. Nanodrugs have the potential to co-deliver various agents in a tissue- and even cell type-specific manner, thereby reducing toxicity and other unwanted side effects associated with non-specific uptake of conventional therapeutics [37]. Therefore, both with regard to nanodrugs designed for the treatment of liver diseases [38] as well as nano-formulations developed to address other organs and cell types, it is highly important to address potential uptake by the liver.

In this regard, most studies have demonstrated endocytic uptake of nanodrugs by KCs [39,40,41,42]. Moreover, we and others have reported that DCs and LSECs may also significantly contribute in this regard [36]. The overall T cell tolerance-promoting function of liver NPCs may even be detrimental in cases of nano-vaccines developed to address antigen-presenting cells such as DCs in secondary lymphoid organs aimed at inducing tumor antigen-specific immune responses [9]. Therefore, it is of general interest to study in detail the interaction of liver NPCs with biologicals in vitro to test and eventually optimize their cell-targeting properties for subsequent in vivo studies. Along this line, ex vivo analysis of liver NPCs at the end of in vivo experiments enables the evaluation of in vivo cell type-targeting behavior and payload-dependent functional effects.

So far, liver perfusion (LP) is the standard method for NPC enrichment but requires laborious in situ pretreatment of the organ with collagenase and DNase I [43,44,45,46]. Hence, this approach limits the number of samples that can be processed. This issue may be especially important when liver NPCs need to be isolated from a number of animals after in vivo treatment for ex vivo analysis. Therefore, we compared the suitability of LP with ex vivo liver dissociation (LD) using a commercially available kit that allows for the parallel processing of numerous samples. We showed that both liver NPC enrichment procedures yielded similar results in terms of overall cell numbers, composition, and viability when assessed directly after isolation. However, after overnight cultivation, NPCs enriched by LD displayed a higher state of activation and were less responsive towards stimulation with different toll-like receptor (TLR) ligands. Altogether, our results show that both methods are suitable for liver NPC isolation. LD proved less laborious than LP but might be more suitable in cases of direct ex vivo analysis of freshly isolated samples, whereas LP is the method of choice when liver NPCs are to be subjected to functional studies in vitro.

## 2. Results

### 2.1. LP and LD Yield Comparable Numbers of Enriched Viable NPCs

LP-mediated NPC enrichment starts with in situ perfusion of the liver via the portal vein with a collagenase/DNase I digestion mixture (Figure 1, left panel), whereas in cases of LD, only extra-corporal enzymatic tissue digestion of liver tissue is performed (Figure 1, right panel). Hence, LP is more demanding and laborious compared to LD. We modified the LD protocol by performing subsequent density centrifugation to enrich NPCs in the interphase (Figure 1, lower panel). This step is part of the LP method by default.

Both liver NPC enrichment methods resulted in similar outcomes in terms of total yield (Figure 2A), the frequency of enriched CD45^+^ liver NPCs (50–60%; Figure 2B) and the composition of the NPC population with regard to F4/80^+^ macrophages (MACs), CD11c^+^ DCs, and CD32b^+^ LSECs (Figure 2C). However, it is worth mentioning that after density centrifugation, cell suspensions obtained by LD contained more debris than observed when performing LP (not shown).

Next, we investigated the viability of enriched liver NPC populations. To this end, we employed FVD, which binds intracellular primary amines of dead cells [47]. In contrast to other live/dead markers, FVD-treated samples can be fixated, which, in contrast to DNA-intercalating agents (see below), allows flow cytometric analysis at later time points.

Unexpectedly, LP-enriched MACs displayed rather strong interactions with the FVD, which suggested that their viability was about 42% only as compared to about 80% in the case of LD-enriched MACs (Figure 2D, left panel). After overnight incubation, MACs interacted with the FVD to a high extent, irrespective of the liver NPC enrichment method—suggesting viability rates below 40%. On the contrary, incubation of liver NPCs both directly after isolation as well as after overnight cultivation with Sytox, which intercalates the genomic DNA of dead cells [48], in general delineated >75% of MACs as viable. Freshly isolated DCs (Figure 2D, middle panel) and LSECs (Figure 2D, right panel) displayed similar FVD/Sytox binding patterns, showing moderately lower viability when enriched by LP and incubated with FVD, whereas no difference in viability was apparent in the case of Sytox incubation. These method-dependent differences were less obvious after overnight incubation, suggesting largely comparable viability rates determined by either viability dye. Similar results were obtained when using FVD conjugated with a distinct fluorochrome, and 7-AAD instead of Sytox (not shown).

Taken together, these observations suggest that LP-mediated enrichment of liver NPCs resulted in stronger interactions with the FVD than after LD enrichment (MACs > DCs, LSECs).

The unexpected finding of enhanced engagement of the FVD by MACs prompted us to perform a comparative viability analysis of spleen cells after isolation as well as after overnight incubation using FVD and Sytox in parallel (Appendix A). None of the freshly isolated splenic immune cell types showed stronger engagement of the FVD than of Sytox. However, after overnight culture B cells, MACs and PMNs displayed a lower viability rate when using FVD for assessment as compared to Sytox. Comparable findings were made when using FVD with a distinct fluorochrome and 7-AAD instead of Sytox (not shown).

Due to the cardinal advantage of FVD over Sytox/7-AAD to fixate samples after treatment, enabling flow cytometric measurements at later time points, we also tested whether Annexin V—which detects phosphatidyl serin on the surface of early/late apoptotic cells [49] and allows for sample fixation as well—could be used instead of FVD. However, a considerable proportion of LD-enriched liver NPCs engaged Annexin V to a similar extent as that observed for FVD (Appendix A).

Altogether, these observations suggest that DNA-intercalating agents (Sytox, 7-AAD) are more reliable for the viability analysis of liver NPCs.

### 2.2. Liver NPCs Enriched by LD Express Costimulatory Receptors at Higher Levels and Are Partially Refractory towards Stimulation

Our findings on the differential binding of liver NPCs to FVD viability dyes suggested that both liver NPC enrichment procedures may also affect their overall state of activation. Overnight-cultivated liver NPC populations displayed no method-associated differences in expression of the antigen-presenting receptor MHCII (Figure 3, Ctrl groups). In contrast, MACs, DCs and LSECs expressed the costimulatory receptors CD80 and CD86 to a higher extent when enriched by LD as compared to LP.

To assess the suitability of LP- and LD-derived liver NPCs for in vitro testing of adjuvants, e.g., as a component of nano-vaccines, we applied several TLR ligands—including the TLR4 ligand LPS, the TLR7/8 agonist R848, native mRNA (TLR3 agonist), the well-established liposomal carrier DOTAP, and DOTAP/mRNA complexes. MHCII expression of MACs enriched by either of the methods displayed was not altered by either stimulus. Similar observations were made for the co-stimulator CD80. In contrast, CD86 expression was upregulated in the case of LP-enriched MACs in response to either stimulus, whereas LD-derived MACs remained unresponsive in this regard. LP-enriched DCs upregulated MHCII and showed similar tendencies regarding both co-stimulators in response to stimulation, whereas DCs pre-activated due to enrichment by LD were refractory.

LP-enriched LSECs showed increased MHCII expression in response to LPS, R848 and DOTAP, but LD-enriched LSECs displayed only very moderately elevated levels. Similar to MACs and DCs, LD-derived LSECs in general scarcely upregulated either co-stimulator in response to stimulation. In contrast, LP-derived LSECs showed somewhat increased CD80 expression when stimulated with R848 and significant upregulation of CD86 upon application of most stimuli.

In line with the partially activated state of LD-enriched liver NPC populations after overnight cultivation, the respective cell culture supernatants contained higher concentrations of TNF-α, IL-6, and IL-10 than in the case of LP-derived NPCs (Figure 4). Type I (IFN-α, -β) and type II (IFN-γ) interferons were scarcely detectable under basal conditions. Most cytokines, except for IFN-α, were upregulated to the highest extent by stimulation with R848 and to a similar extent, in an enrichment method-independent manner. Only IL-10 showed markedly higher absolute levels in the case of LD-enriched NPCs than those observed for LP-derived NPCs. Stimulation with LPS also increased cytokine concentrations—again except for IFN-α—albeit to a lower extent than that induced by R848. Again, for some cytokines (TNF-α, IL-6, IL-10), levels were somewhat higher in the case of LD-derived NPCs. Only in the case of IFN-γ did LP-derived NPCs produce more cytokine than that observed for LD-enriched NPCs. mRNA and DOTAP, when applied separately, exerted no stimulatory effect on the production of either cytokine—except for IFN-γ, which was markedly upregulated in the case of LP-enriched NPCs treated with mRNA. DOTAP/mRNA complexes, however, specifically increased type I IFN concentrations—again, interestingly, to a higher extent in the case of LP- than LD-derived liver NPCs.

Altogether, these results suggest that LD results in the partial activation of enriched liver NPCs, which in the case of surface activation markers prevented stimulation-dependent upregulation. In agreement, LD-enriched liver NPCs showed higher basal levels of some cytokines, but in contrast to inducing non-responsiveness on the level of surface markers, were capable of upregulating cytokine production in response to stimulation.

## 3. Discussion

The immunological role of liver NPCs, and the frequent observation that nanodrugs intended to address other cell types are frequently enriched in this organ, underline the importance of studying the interactions of liver NPC populations with nanodrugs and the functional consequences thereof. Furthermore, liver NPCs have also been recognized as an interesting target population for nanodrug-mediated approaches aimed at exploiting their intrinsic protolerogenic potential to inhibit unwanted immune reactions, as in the case of allergies [50]. Moreover, in the case of anti-tumor nano-vaccination approaches, it may be an important issue to overcome their tolerance-promoting properties by applying suitable adjuvants, as assessed in this study [9].

Therefore, the suitability of liver NPC isolation methods is an important issue. In this regard, LP has been well established over the last decades, and numerous modifications have been tested—as summarized in Table 1. However, LP requires time-consuming in situ treatment of the liver, which may limit the number of samples that can be processed, e.g., for ex vivo analysis after in vivo treatments—as for example in the case of biodistribution studies of nano-carriers [36]. Therefore, we compared LP with LD as an alternative approach in this study.

We show that both enrichment methods yielded overall comparable results in terms of yield of CD45^+^ liver NPCs [9], displaying similar compositions of NPC populations with regard to DCs, MACs, and LSECs. Surprisingly, however, we observed that LD-enriched liver NPCs, and MACs especially, showed unspecific interactions with the FVD, and thereby would be considered dead. Similar findings were also made for the other liver NPCs populations. Furthermore, both LP- and LD-enriched MACs displayed low viability after overnight cultivation when using FVD. In contrast, DNA-intercalating agents (Sytox, 7-AAD) identified only <25% of MACs as dead, even after overnight incubation. Besides this, splenic MACs and PMN were also characterized by stronger interaction with FVD than DNA-intercalating dyes after overnight cultivation. Therefore, these findings suggest that endocytically active cell types in a viable state are prone to internalize FVD, which favors the use of DNA-binding agents for delineating viability. It is unclear as of yet which receptors may play a role in this regard.

However, LD enrichment also resulted in partial activation of the assessed liver NPC types, which may compromise the outcome of in vitro studies aimed, e.g., at assessing the stimulatory activity of agents—as exemplified in this study for several TLR ligands. In this regard, the responsiveness of LD-dependently pre-activated liver NPCs on the level of surface markers was strongly impaired. LD-enriched NPCs generated easily detectable amounts of TNF-α, IL-6, and IL-10—even in the absence of stimulatory agents—but the difference towards LP-derived NPCs was most apparent in the case of IL-10. This potent anti-inflammatory cytokine has been implicated as an effector molecule upregulated in response to different types of stress [64] and has been shown to contribute in an essential manner to the overall pro-tolerogenic role of liver NPCs [65]. KCs have been reported to express IL-10 receptor at high levels, resulting in sustained activation of the transcription factor signal transducer and activator of transcription (STAT)3 [66]. STAT3 in turn is a major driver of IL-10 expression, thereby establishing a positive feedback loop for this cytokine [67]. However, LD-derived liver NPCs were still largely responsive towards stimulation on the cytokine level. Here, we observed that R848, and to lesser extent LPS, was a potent stimulator of TNF-α, IL-6, IL-10, and IFN-γ production—as previously shown by us and others [68]. In addition, we also demonstrated that complexed mRNA yielded considerable induction of type I IFN, as has also been described for antigen-presenting cells in secondary lymphoid organs upon transfection with mRNA-delivering liposomes [69,70,71,72,73].

Our results show that LP-derived liver NPCs can be used to study, e.g., the stimulatory effects of adjuvant-loaded nanodrugs in terms of surface activation marker expression and cytokine production. With regards to the latter, it will also be interesting to assess cytokine production on the single cell level by intracellular cytokine detection [74]. In the case of fluorescence-labeled nano-carriers, it will also be possible to study their liver NPC type-specific engagement—especially in the case of nanodrugs decorated with, e.g., C-type lectin receptor-targeting moieties such as carbohydrates [75]. Finally, in the case of nano-vaccines that co-deliver an antigen, subsequent cocultures of accordingly pretreated liver NPCs with antigen-specific T cells will show whether adjuvant-mediated NPC activation is able to overcome their intrinsic protolerogenic properties and result in efficient activation of antigen-specific T cells, e.g., to induce profound anti-tumor T cell responses [76].

Altogether, we conclude that LP is more reliable than LD for obtaining liver NPCs for subsequent in vitro studies. However, LP represents a laborious technique and well-trained operators are needed (Table 2). Therefore, LD is the method of choice in the case liver NPCs that need to be derived from a larger number of in vivo-treated animals for subsequent ex vivo analysis. In this case, the potential of LD to process a larger number of samples in parallel enables the completion of NPC isolation within a shorter period of time as compared to the rather continuous mode of handling in the case of LP. Additionally, our finding that LD partially activates NPCs in the course of subsequent overnight culture is not of primary concern when samples are to be analyzed directly after NPC isolation.

## 4. Materials and Methods

### 4.1. Animals

C57BL/6 mice were bred and housed in the Translational Animal Research Center (TARC) of the Johannes Gutenberg-University Mainz under specific pathogen-free conditions on a standard diet, according to the guidelines of the regional animal care committee. The guide for the care and use of laboratory animals [77] as well as the 3R principles in laboratory animal experiments [78] were followed. Mice were sacrificed at an age of between 12–16 weeks for organ retrieval according to § 4(3) TierSchG.

### 4.2. Spleen Cells

Spleens were mashed using a sterile syringe plunger on a PBS pre-soaked 40 µm cell strainer (EASYstrainer™; Sarstedt, Nümbrecht, Germany). Mashed spleen tissue was washed through the strainer using 10 mL of PBS/2 mM EDTA buffer, and cells were pelleted by centrifugation (1200 rpm, 10 min, 4 °C).

### 4.3. Enrichment of Liver NPCs

Mice NPCs were enriched by liver perfusion (LP) and liver dissociation (LD). For LP, mice were anesthetized with a Ketamine/Xylazine mixture (Ketamine 120 mg/Kg; Xylasin 16 mg/kg) and the abdominal cavity was opened (see Figure 1, left panel). The vena portae was cannulated and flushed with 20 mL of HBSS (Hank’s Balanced Salt Solution; ThermoFisher, Waltham, MA, USA) containing 100 U/mL of collagenase A (Sigma-Aldrich, St. Louis, MO, USA) and 10 µg/mL of DNase I (Sigma-Aldrich, St. Louis, MO, USA). Then, the liver was retrieved, cut into pieces, and incubated for 15 min at 37 °C in a 50 mL tube with PBS containing 100 U/mL of collagenase A and 10 µg/mL of DNase I. Afterwards, the liver tissue was mashed through a 70 µm nylon cell strainer and medium (DMEM (Dulbecco’s Modified Eagle Medium)/F-12 GlutaMAX™, ThermoFisher Scientific) was applied. The cell suspension was subjected to density centrifugation using 30% Histodenz-HBSS (both from Sigma-Aldrich, St. Louis, MO, USA) gradient centrifugation as described [45] (see Figure 1, lower panel).

For LD-mediated NPC enrichment, mice were killed by cervical dislocation and the liver was retrieved (see Figure 1, right panel). An enzyme-dependent dissociation mixture of proprietary composition (Liver dissociation kit; Miltenyi Biotec, Bergisch-Gladbach, Germany) was preincubated for 15 min in C tubes (Miltenyi Biotec). Then, liver tissues were cut in little pieces and transferred into prepared C tubes. The latter were placed into a gentleMACS Dissociator and the tissue was minced (two times program m_liver_03). Afterwards, the cell suspension was incubated for 30 min at 37 °C under continuous shaking, followed by another round of gentleMACS-mediated disintegration (two times program m_liver_04). The samples were cleared using a Cell Strainer (100 µm; Sarstedt) and the liver NPCs were enriched by density centrifugation (see above).

### 4.4. Stimulation of Liver NPCs

Liver NPCs were seeded into wells of 96-well plates (10^5^/100 µL) and were treated overnight with LPS, (100 ng/m; Kenilworth, NJ, USA), R848 (1 µg/m, Ínvivogen, San Diego, CA, USA), ovalbumine-encoding mRNA (1 µg/mL; Tri-Link, San Diego, CA, USA), DOTAP (6.4 µL/mL; Roth, Weiden, Germany), and DOTAP/mRNA complexes, which were formed by mixing both components at the mentioned amounts as recommended by the manufacturer (Roth).

### 4.5. Flow Cytometry

Single cell suspensions were washed (2% FCS in PBS), and Fc receptors were blocked with aCD16/CD32 (clone 2.4G2 clone; Biolegend, San Diego, CA, USA) for 10 min. Afterwards, cells were incubated with receptor-specific antibodies (Appendix A) obtained from ThermoFisher (Waltham, MA) or BD Biosciences (Franklin Lakes, NJ, USA) for 20 min at 4 °C. Afterwards, cell viability was assessed by using fixable viability dye (FVD)-450 and -780, 7AAD, Sytox AADvanced (all from ThermoFisher) and Annexin V-AF647 (Biolegend), as recommended by the manufacturer. Samples were subjected to flow cytometry using a Attune NxT Flow Cytometer (Thermo Scientific) and an LSRII (BD Biosciences), and analyzed with Attune NxT ((ThermoScientific and FlowJo software, respectively (BD Biosciences)). The gating strategies for liver NPCs (see Appendix A) and splenic leukocytes (see Appendix A) are depicted in the supplementals.

### 4.6. Cytokines

Supernatants of cell cultures were retrieved and stored at −20 °C for subsequent analysis. TNF-α, IFN-γ, IL-6, IL-1ß, and IL-10 were quantified using a Cytometric Bead Assay (CBA; BD Biosciences) as recommended by the manufacturer. Results were analyzed using FCAP Array Analysis Software v.1.0.1 (BD Biosciences). IFN-α and IFN-β were measured using Legendplex reagents as recommended and data were analyzed using Qognit Legendplex Analysis Software (both from Biolegend).

### 4.7. Data Analysis

Data were analyzed using GraphPad PRISM v5 and v9 software (GraphPad Software Inc., San Diego, CA, USA). Data are given as mean ± SEM for the indicated number of independent experiments. Statistical differences were determined by unpaired Student’s *t*-tests when comparing two groups, one way ANOVA followed by post-hoc Tukey‘s test when comparing multiple groups monitored at a given time point, and two-way ANOVA/Tukey tests when comparing multiple groups assessed at different time points—assuming significant differences at *p* < 0.05 in either case.

## Figures and Tables

**Figure 1 ijms-23-06543-f001:**
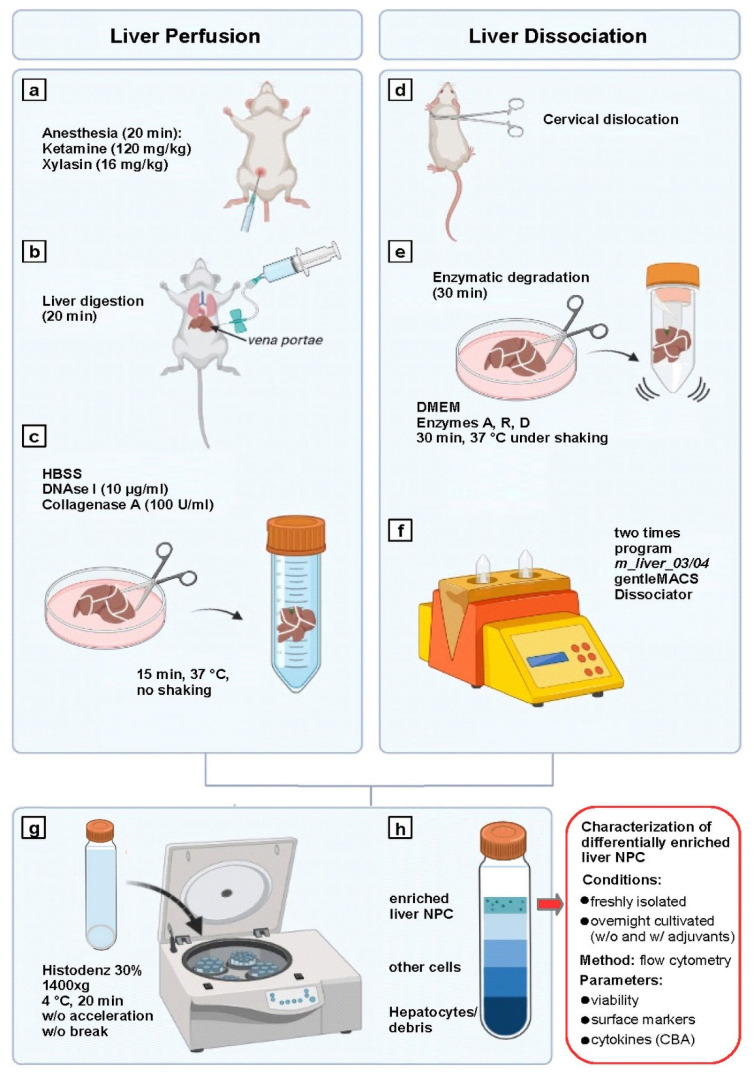
Schematic overview of liver non-parenchymal cell (NPC) enrichment by liver perfusion (LP) and liver dissociation (LD) and subsequent experimental readout. Left panel: LP mice were (**a**) anesthetized with ketamine/Xylazine, (**b**) the vena portae was cannulated and collagenase A/DNase I-containing PBS was infused for in situ tissue digestion. (**c**) The liver was retrieved, cut into pieces and further digested with collagenase A/DNase I. Liver tissue pieces were mashed to obtain a single cell suspension (not shown). Right panel: LD mice were (**d**) killed by cervical dislocation, and (**e**) the retrieved liver was cut into pieces and digested in an enzyme cocktail of proprietary composition (Miltenyi Biotec), (**f**) followed by mechanical dissociation using a gentleMACS (Miltenyi Biotec). Lower left panel: Liver cell suspensions generated by either method were (**g**) subjected to Histodenz density gradient centrifugation, (**h**) resulting in enrichment of liver NPCs in the interphase that was subjected to subsequent experiments. Lower right panel: differentially enriched liver NPCs were characterized in a freshly isolated state and after overnight incubation w/o and in the presence of adjuvants with regard to viability, surface marker expression, and cytokine contents by flow cytometry. (created with BioRender.com; accessed on 1 June 2022).

**Figure 2 ijms-23-06543-f002:**
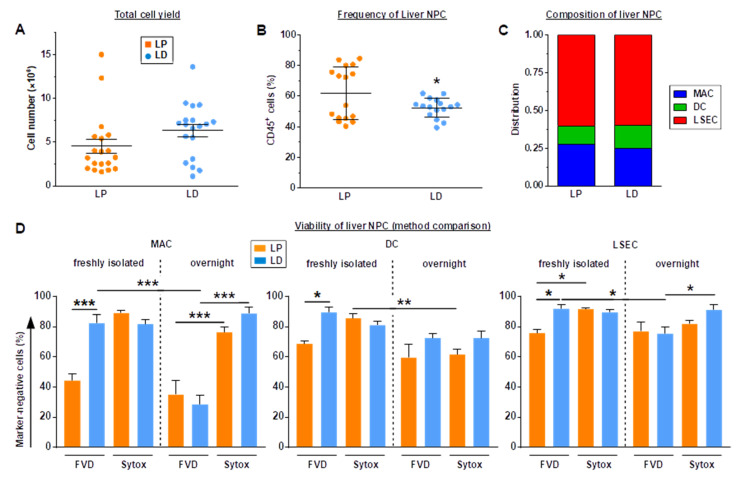
LP and LD yield comparable numbers and composition of liver NPCs, but LD-enriched cells appear dead when using FVD. (**A**) Total cell yield obtained after density centrifugation (mean ± SEM, each *n* = 19). (**B**) Frequencies of CD45^+^ cells within the enriched NPC suspensions were assessed by flow cytometry (mean ± SEM, each *n* = 19). (**C**) Composition of liver NPCs (CD11c^+^ DCs, F4/80^+^ MACs, CD32b^+^ LSECs) within the CD45^+^ cell population (mean, each *n* = 19; the gating strategy is shown in Appendix A). Statistically significant differences versus (**B**) *LP (*t*-test) and (**D**) * between groups (two-way ANOVA/Tukey test) are indicated. * *p* < 0.05, ** *p* < 0.01, *** *p* < 0.001.

**Figure 3 ijms-23-06543-f003:**
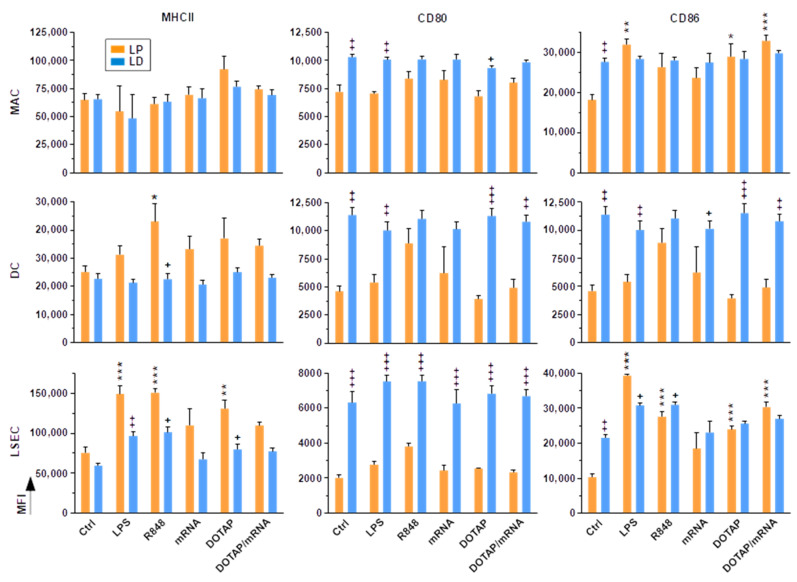
LD-enriched liver NPCs are pre-activated, as deduced from elevated activation marker expression. Liver NPCs enriched by LP and LD were seeded into wells of 96 well plates (10^5^/100 µL) and stimulated overnight with the agents indicated or left untreated. The man fluorescence intensity (MFI) of surface markers was assessed by flow cytometry (mean ± SEM, *n* = 4). Statistically significant differences versus * according Ctrl and versus ^+^LP are indicated (one way ANOVA, Tukey test). *^, +^ *p* < 0.05, **^, ++^ *p* < 0.01, ***^, +++^ *p* < 0.001.

**Figure 4 ijms-23-06543-f004:**
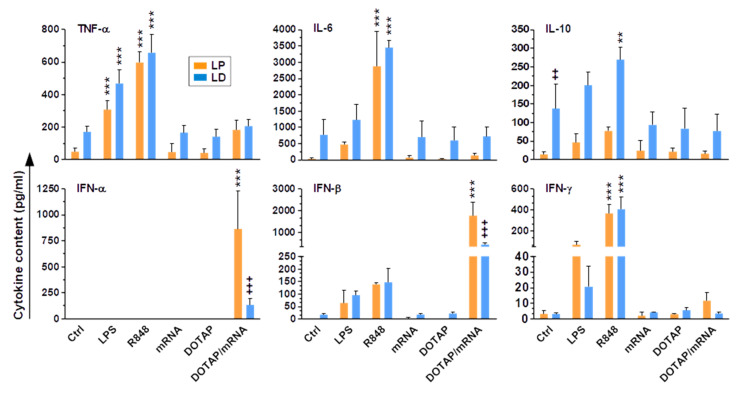
Liver NPCs generate cytokines in an adjuvant-specific manner and depending on the enrichment method used. Supernatants of differentially stimulated liver NPCs (see Figure 3) were assayed for cytokine contents in culture supernatants (mean ± SEM, *n* = 3). Statistically significant differences versus * according Ctrl and versus ^+^LP are indicated (one way ANOVA, Tukey test). **^, ++^ *p* < 0.01, ***^, +++^ *p* < 0.001.

**Table 1 ijms-23-06543-t001:** Protocols for liver NPC enrichment.

Liver Digestion	Access to the Liver	Species	NPC Yield (×10^6^)	Gradient Centrifugation	Viability (%) ^1^	Reference
Perfusion (aeration apparatus)	Vp	r	-	-	-	[51]
0.05% Collagenase/0.10% hyaluronidase	Vp	r	-	-	-	[52]
Pronase (*Streptomyces griseus*)	Vp	r	2–15	yes	-	[53]
Pronase (*Streptomyces griseus*)	-	r	14/gr	-	-	[54]
Collagenase	-	r	0.5–0.6/gr	Percoll	-	[55]
Pronase/DNAse/Collagenase	Vp	r	26 (LSEC), 13 (KC)	Stractan	90	[56]
Collagenase	Vp	m	9	Metrizamide	-	[57]
Collagenase/DNAse	Vp	m	-	Histodenz	-	[43]
Collagenase/DNAse	Vp	m	-	Histodenz	-	[44]
Collagenase IV/Pronase	Vp	m	-	Percoll	97	[9]
Collagenase IV (*C. histolyticum*)	ex vivo	h	KC: 1.8, LSEC: 4.3/gr	yes	-	[58]
EGTA/collagenase P	ex vivo	h	1.9 (KC), 0.27 (LSEC)/gr		>90	[59]
According to [53]	Vp	m	-	yes	-	[60]
Collagenase II	ex vivo	m	5/gr tissue	-	-	[61]
Collagenase I	Vp	r	KC: 3–5, LSEC: 18–20	Percoll	-	[62]
EGTA/collagenase P (two-step)	ex vivo	h	-	Yes	-	[63]
Collagenase/DNAse	Vp	m	3.33	Histodenz	-	[36]

h: human, KC: Kupffer cell; LSEC: liver sinusoid endothelial cell; m: murin, Vp: vena portae. ^1^ numbers given when viability was assessed in the study.

**Table 2 ijms-23-06543-t002:** Advantages and disadvantages of using LP or LD for NPC enrichment.

Aspect		LP	LD
Total yield		+++	+++
Purity (CD45^+^)		62%	52%
NPC viability ^1^	MAC	89% ^2^, 76% ^3^	82%, 89%
	DC	86%, 62%	81%, 72%
	LSEC	92%, 82%	90%, 91%
NPC activation state		o/+	+++
NPC responsiveness towards stimulation		++	o/+
Preparation time		4–5 h	2–3 h
Costs		+	+++

^1^ given for Sytox; ^2^ freshly isolated; ^3^ after overnight culture. O, unaltered; +, low; ++, intermediate, +++, high.

## Data Availability

Not applicable.

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
