# Peer review of "Enrichment Methods for Murine Liver Non-Parenchymal Cells Differentially Affect Their Immunophenotype and Responsiveness towards Stimulation"

_ijms, 2022, doi:10.3390/ijms23126543_

Round 1

Reviewer 1 Report

Comment:

This paper discusses " Enrichment methods for murine liver non-parenchymal cells differentially affect their immunophenotype and responsiveness towards stimulation ". The main contribution of the paper is " it compares the suitability of two isolation methods for murine liver NPC. Liver perfusion (LP) with collagenase/DNase applied via portal vein leads to efficient liver digestion, whereas the modified liver dissociation (LD) method combines mechanical dissociation of the retrieved organ with enzymatic degradation of the extracellular matrix"

This is an interesting study and is generally well written and structured. However, in my opinion the paper has some shortcomings in regards to the recent references which are recommended.

Minor comments:

·         Well written except in some situations. I advise recheck it again.

·         The introduction should be advised to be re-written to be in more logical flow.

·         How many animals used for each experiments?

·         The methods in details should be described and analysis as well .

·         Please, Suggest future experiments in details

·         Please, Specify the most specific protein from liver that you suggest might be related to this topic.

·         Please, try to add general paragraph about importance of liver cells. Why you studied this specific cells.

·         Figure 1 and 2 are not obvious to me. It is not understandable.

·         Figure 1 and 2 are too small.

·         Although it needs to be in more logical flow, the introduction provides a good, generalized background of the topic. However, why not cite more literature papers .

·         I think the motivations for this study need to be made clearer

·         Regarding the figures: I recommend make more figures to be illustrative.

Gi

Author Response

This paper discusses " Enrichment methods for murine liver non-parenchymal cells differentially affect their immunophenotype and responsiveness towards stimulation ". The main contribution of the paper is " it compares the suitability of two isolation methods for murine liver NPC. Liver perfusion (LP) with collagenase/DNase applied via portal vein leads to efficient liver digestion, whereas the modified liver dissociation (LD) method combines mechanical dissociation of the retrieved organ with enzymatic degradation of the extracellular matrix"

 This is an interesting study and is generally well written and structured. However, in my opinion the paper has some shortcomings in regards to the recent references which are recommended.

Minor comments:

Well written except in some situations. I advise recheck it again.

We have thorough corrected the manuscript to enhance calrity and readibility.

The introduction should be advised to be re-written to be in more logical flow.

We have restructured and extended parts of the introduction to enhance clarity.

How many animals used for each experiments?

We have included an explanatory statement in the materials and methods section (4.7. Data analysis).

The methods in details should be described and analysis as well.

 The schematic overview presented (new figure 1) was designed to clarify the methodoloical approach both in terms of liver NPC enrichtment and the subsequent experimental workflow.

Please, suggest future experiments in details.

We have incuded an according paragraph in the last part of the discussion.

Please, Specify the most specific protein from liver that you suggest might be related to this topic.

We have extended the introduction to present more clearly the various cell types of the liver, and which function especially hepatocytes fulfill, including the generation of important serum factors like albumin.

Please, try to add general paragraph about importance of liver cells. Why you studied this specific cells.

We have extended the description of the characteristics and functions of the various types of liver cells.

Figure 1 and 2 are not obvious to me. It is not understandable.

We have replaced figure 1 by a schematic overview of the two liver NPC isolation methods to present the methodological approaches in a clearer manner. Further,we have added explanatory headlines to each part of figure 2.

Figure 1 and 2 are too small.

We have increased the size of both figures.

Although it needs to be in more logical flow, the introduction provides a good, generalized background of the topic. However, why not cite more literature papers.

We have revised and extended the introduction in this regard.

I think the motivations for this study need to be made clearer

We have revised the manuscript in the introduction and discussion sections to clarify the aim of the study.

Regarding the figures: I recommend make more figures to be illustrative.

We have included a schematic overview of the liver perfusion and liver dissociation workflows and the subsequent experimental procedures.

Reviewer 2 Report

The article provides an interesting comparison of two different techniques to purify hepatic parenchymal cells. It provides sufficient details on the different cell populations and stimulations. It does not however show any morphological alterations that may occur during the procedures.

It would be useful to the reader that in table 1, the items yield, purity and viability be expressed in % rather than + signs.

The authors should discuss more why IL10 (figure 4) is the only cytokine whose secretion markedly differs between the two separation methods.

Table S1 should be in the main paper since it compares different procedures, although the table should include viability in all cases.

The preparation methods should be performed by anyone following the procedure therefore the preparation effort in the same table does not make sense. 

Author Response

The article provides an interesting comparison of two different techniques to purify hepatic parenchymal cells. It provides sufficient details on the different cell populations and stimulations. It does not however show any morphological alterations that may occur during the procedures.

It would be useful to the reader that in table 1, the items yield, purity and viability be expressed in % rather than + signs.

We have altered the table as suggested by the reviewer.

The authors should discuss more why IL10 (figure 4) is the only cytokine whose secretion markedly differs between the two separation methods.

We address this issue in the discussion.

Table S1 should be in the main paper since it compares different procedures, although the table should include viability in all cases.

We transferred this table to the discussion section (new table 1). Viability frequencies are given (6th row) when mentioned in the according study. (We have added an explanatory footnote to indicate so.

The preparation methods should be performed by anyone following the procedure therefore the preparation effort in the same table does not make sense. 

We have deleted the according line in the table.

Reviewer 3 Report

The study of Medina-Montano et al. aims to compare the suitability of two isolation methods for murine liver NPC, such as liver perfusion (LP) and the modified liver dissociation (LD), finding that the two methods are different regarding the degree of activation after overnight cultivation, and that LD-enriched NPC are less responsive towards stimulation with conventional used activators.

The topic addressed is of interest and the results seem to have scientific soundness. Only a few clarifications would deserve the attention of the authors to better illustrate the context.

The authors declare that liver non parenchymal cells comprise Kupffer cells, dendritic cells and liver sinusoidal endothelial cells. Regarding the control of the immunological state within the liver, this statement could be sufficient but, if we wanted to consider more globally the tissue homeostasis of the liver, among the non-parenchymal cells of the liver we should also consider the hepatic stellate cells. The isolation protocols for hepatic stellate cells are certainly different and probably are not the subject of this paper, however they should be mentioned if it is not possible to include them in the experimental plan, due to their fundamental role in normal and pathological conditions.

Author Response

The study of Medina-Montano et al. aims to compare the suitability of two isolation methods for murine liver NPC, such as liver perfusion (LP) and the modified liver dissociation (LD), finding that the two methods are different regarding the degree of activation after overnight cultivation, and that LD-enriched NPC are less responsive towards stimulation with conventional used activators.

The topic addressed is of interest and the results seem to have scientific soundness. Only a few clarifications would deserve the attention of the authors to better illustrate the context.

The authors declare that liver non parenchymal cells comprise Kupffer cells, dendritic cells and liver sinusoidal endothelial cells. Regarding the control of the immunological state within the liver, this statement could be sufficient but, if we wanted to consider more globally the tissue homeostasis of the liver, among the non-parenchymal cells of the liver we should also consider the hepatic stellate cells. The isolation protocols for hepatic stellate cells are certainly different and probably are not the subject of this paper, however they should be mentioned if it is not possible to include them in the experimental plan, due to their fundamental role in normal and pathological conditions.

We thank the reviewer for this hint. We have extended the introduciton to explain briefly the role of hepatic stellate cells in liver diseases.